# Monitoring of Delayed Cerebral Ischemia in Patients with Subarachnoid Hemorrhage via Near-Infrared Spectroscopy

**DOI:** 10.3390/jcm9051595

**Published:** 2020-05-24

**Authors:** Jeong Jin Park, Chulho Kim, Jin Pyeong Jeon

**Affiliations:** 1Department of Neurology, Konkuk University Medical Center, Seoul 05030, Korea; medicalstory@gmail.com; 2Department of Neurology, Hallym University College of Medicine, Chuncheon 24253, Korea; gumdol52@naver.com; 3Division of Big Data and Artificial Intelligence, Chuncheon Sacred Heart Hospital, Chuncheon 24253, Korea; 4Institute of New Frontier Stroke Research, Hallym University College of Medicine, Chuncheon 24253, Korea; 5Department of Neurosurgery, Hallym University College of Medicine, Chuncheon 24253, Korea; 6Genetic and Research Inc., Chuncheon 24253, Korea

**Keywords:** subarachnoid hemorrhage, delayed cerebral ischemia, near-infrared spectroscopy

## Abstract

We investigated the role of near infrared spectroscopy (NIRS) in identifying delayed cerebral ischemia (DCI) in patients with subarachnoid hemorrhage (SAH). We measured the cerebral regional oxygen saturation (rSO2) continuously for 14 days. The differences in rSO2 according to DCI were analyzed. We also compared the diagnostic accuracy of NIRS and transcranial Doppler ultrasonography (TCD) for DCI detection using the area under receiver operator characteristic (ROC) curve. Fifty-two patients treated with coil embolization were enrolled, including 18 with DCI (34.6%) and 34 without DCI (65.4%). Significant differences in rSO2 levels were observed from days 7 to 9. The rSO2 level was 60.95 (58.10–62.30) at day 7 in the DCI vs. 63.90 (62.50–67.10) in the non-DCI patients. By day 8, it was 59.50 (56.90–64.50) in the DCI vs. 63.30 (59.70–68.70) in the non-DCI cases. By day 9, it was 61.85 (59.40–65.20) in the DCI vs. 66.00 (62.70–68.30) in the non-DCI. A decline of >12.7% in SO2 rate yielded a sensitivity of 94.44% (95% CI: 72.7–99.9%) and a specificity of 70.59% (95% CI: 52.5–84.9%) for identifying DCI. Changes in NIRS tended to yield better diagnostic accuracy than TCD, but were not statistically significant. NIRS is a feasible method for real-time detection of DCI.

## 1. Introduction

Delayed cerebral ischemia (DCI) is a well-known risk factor for serious neurologic complications following subarachnoid hemorrhage (SAH) [1]. Cerebral vasospasm has been regarded as an important cause of DCI, as well as its therapeutic target [2]. In addition to regular neurological examination, transcranial doppler ultrasonography (TCD) is the main tool used for DCI monitoring. Nevertheless, operator dependency, anatomical difficulty, and a weak correlation between elevated blood flow velocity and symptomatic vasospasm are challenges that are still unresolved. Therefore, a simple noninvasive, robust, and continuous monitoring method for bedside assessment of cerebral ischemia status is required for patients with SAH who are at high risk for DCI. 

Near-infrared spectroscopy (NIRS) is a technique that has been used to detect real-time cortical hemodynamic status of oxyhemoglobin and deoxyhemoglobin by employing light wavelengths in the near-infrared spectrum [3]. NIRS has been widely used to monitor brain oxygenation during cardiac or carotid surgery [3]. In piglets, NIRS was used to predict histologic brain damage and high-energy phosphates [4]. In general, a reduction in cerebral regional oxygen saturation (rSO2) greater than 20% or levels below 50–60% is associated with hypoxic or ischemic injury [5,6]. Levy et al. [6] reported that the ischemic threshold of rSO2 was 47% in 10 patients during ventricular fibrillation. Nevertheless, the results of previous studies cannot be used to develop clinically useful cutoff points to identify patients with DCI, since most of the results were based on pediatric cases and patients without cerebrovascular disease [5]. Therefore, this study was performed to investigate the clinical feasibility of NIRS and its optimal threshold for the detection of DCI following SAH.

## 2. Materials and Methods 

### 2.1. Study Population

The study cohort was derived from the Chuncheon Sacred Heart Hospital stroke database between September 2016 and March 2019. This database is an ongoing prospective study conducted at the regional medical center of the district of Chuncheon City, the capital city of Gangwon Province in Korea [7,8,9,10,11,12]. We included SAH patients from this database, who were screened for DCI using an NIRS device based on in vivo optical spectroscopy (INVOS) system (INVOS, Somanetics Corp, Troy, MI, USA). The inclusion criteria were (1) spontaneous SAH in patients >18 years old; and (2) patients who underwent coil embolization to secure the aneurysm rupture. The exclusion criteria were (1) non-spontaneous SAHs due to trauma or infarction; and (2) patients who underwent aneurysm clipping. Because SAH patients with surgical clipping had air, fluid, or hemorrhage on the craniotomy sites, which can distort NIRS measurement, we only placed the NIRS sensor pad in patients treated with coil embolization.

### 2.2. Patient Monitoring and Outcomes

Cerebral rSO2 was monitored electronically every two seconds using an INVOS system with two probes placed on the upper forehead according to the manufacture’s protocol [13]. The data were recorded and analyzed in the ICU workstation. The primary outcome was the association between NIRS and DCI following SAH, suggesting the optimal cutoff value of rSO2 reduction for identification of DCI. The secondary endpoint was the comparative diagnostic accuracy of NIRS and TCD to determine the role of severe vasospasm in DCI associated with cerebral angiography as the reference value. 

After coil embolization, we conducted CT angiography on days 3, 7, and 14 routinely to assess the degree of vasospasm after surgery. After excluding possible causes, such as rebleeding, hydrocephalus, seizures, or electrical abnormalities, DCI was diagnosed by the following conditions: (1) new neurologic changes, including dysphasia, motor weakness, sensory change, and decreased consciousness, or decreased GCS score of at least 2 points; and (2) severe cerebral vasospasm [14]. When DCI was suspected, cerebral angiography was performed to evaluate the degree of vasospasm warranting chemical angioplasty. 

Severe vasospasm found on TCD was defined by a mean flow velocity higher than 200 cm/s in the middle cerebral artery (MCA) or 85 cm/s in the basilar artery [15,16]. To prevent DCI due to vasospasm, nimodipine (20 μg/kg/h; Samjin Pharmaceuticals, Seoul, Korea) was administered intravenously within 24 h after ictus [17]. Medical information (age, blood pressure, heart rate, hypertension (HTN), diabetes mellitus (DM), hyperlipidemia, smoking, and hemoglobin) and radiological findings (Hunt and Hess (H-H) grade, Fisher grade, aneurysm location, and size) were independently reviewed by two neurointerventionists (J.P.J. and J.J.P.). Any disagreement was resolved by a third reader. This study was approved by the Institutional Review Board of the participating hospital (No. 2016-3, 2017-9, 2018-6, and 2019-06).

### 2.3. Statistical Analysis

Descriptive analyses are presented as the numbers of subjects (percentage) for categorical variables, and the means and standard deviations (SD) are shown. The rSO2 changes of patients with and without DCI were compared using the Mann-Whitney U test. A receiver operator characteristic (ROC) curve was generated to determine the optimal cutoff value for diagnosing DCI [18,19]. The predictive value of severe vasospasm contributing to DCI was determined using the area under ROC curve (AUROC) between NIRS and TCD. Linear mixed models were developed to assess the association of the rSO2 with the DCI, because rSO2 values were measured repetitively in the same subjects. In this model, H-H grade, Fisher grade, mean arterial pressure (MAP), DCI status, and the timing (days 1 to 14; intra-subject variable) were set as fixed effects. All statistical analyses were performed using SPSS V.21 (SPSS, Chicago, IL, USA) and MedCalc (www.Medcalc.org) with statistical significance indicated at *p* < 0.05.

## 3. Results 

### 3.1. Clinical Characteristics of the Enrolled Patients

The baseline characteristics of the enrolled patients are shown in Table 1. Among 52 SAH patients, 18 (34.6%) experienced DCI during the follow-up period. There were no statistical differences involving female gender, age, HTN, DM, hyperlipidemia, smoking, heart rate, or hemoglobin between the two groups. Most aneurysms (86.5%) were located in the anterior circulation. The patients with DCI tended to manifest poorer clinical outcomes than those without DCI. Twenty-five of the patients were intubated after coil embolization, and 11 patients were diagnosed with DCI.

### 3.2. Measurement of rSO2 According to DCI

We measured rSO2 consecutively for 14 days after ictus. The rSO2 results were similar for three days after ictus, followed by a decline in the rSO2 levels of patients with DCI compared with those without DCI. Specifically, patients with DCI showed a marked decline in rSO2 on days 7 to 9. By day 7, the rSO2 value was 60.95 (58.10–62.30) in the DCI vs. 63.90 (62.50–67.10) in non-DCI patients (*p* = 0.001). By day 8, it was 59.50 (56.90–64.50) in the DCI vs. 63.30 (59.70–68.70) in non-DCI cases (*p* = 0.015) and on day 9, it was 61.85 (59.40–65.20) in the DCI vs. 66.00 (62.70–68.30) in the non-DCI patients (*p* = 0.005). We further evaluated the association between rSO2 levels and the onset of DCI. Patients presented with varying onsets of DCI: 5 days, *n* = 1 (5.6%); 6 days, *n* = 2 (11.1%); 7 days, *n* = 6 (33.3%); 8 days, *n* = 5 (27.7%); 9 days, *n* = 2 (11.1%); 10 days, *n* = 1 (5.6%); and 11 days, *n* = 1 (5.6%) (Figure 1).

Based on the decline in rSO2 with the laboratory results on day 1 as the reference standard, we created an ROC curve (AUC = 0.865, 95% CI: 0.742–0.944). Rates of rSO2 decrease >12.7% yielded the most favorable balance of test characteristics, a sensitivity of 94.44% (95% CI: 72.7–99.9%) and a specificity of 70.59% (95% CI: 52.5–84.9%) (Figure 2).

As shown in Table 2, the results of the linear mixed-effects model of predictors of rSO2 according to DCI suggest that a high H-H grade was associated with decreased rSO2. Fisher grade or MAP was not associated with rSO2 levels. However, rSO2 in patients with DCI was 1.46% lower than in non-DCI patients (*p* = 0.004).

### 3.3. Diagnostic Comparison of NIRS and TCD Velocity

We further evaluated the diagnostic accuracy of NIRS changes (>12.7%) compared with TCD in patients with severe angiographic vasospasm progressing to DCI. Six out of 52 SAH patients (11.5%) were excluded from the diagnostic comparison due to poor bone window. Accordingly, 46 patients were finally included in the analysis. In severe angiographic vasospasm, TCD showed a sensitivity of 75.0% (95% CI: 47.6–92.7%), specificity of 80.0% (61.4–92.3%), and a negative predictive value of 85.7% (71.6–93.5%). As shown in Figure 3, NIRS changes appear to be associated with better diagnostic accuracy than TCD with a difference between areas of 0.044 (95% CI: −0.108–0.195), without any statistical significance (Figure 3).

## 4. Discussion 

This study showed that rSO2 observed on NIRS allowed for the real-time monitoring of DCI development. The optimal cutoff rate of rSO2 reduction for identifying DCI was 12.7%, which had a sensitivity of 94.44% and a specificity of 70.59%. 

The clinical utility of NIRS for DCI detection is disputed. Naidech et al. [20] reported that rSO2 was not associated with angiographic or TCD vasospasm in six patients. In contrast, Rothoerl et al. [21] demonstrated a high correlation rate of 90% between NIRS and tissue pO2 measurements. Yokose et al. [22] also reported that NIRS was better than TCD for DCI detection. In their study, NIRS was successfully used to identify three DCI patients, although the TCD velocity of one patient remained within the normal range. We believe that the difference in treatment methods led to conflicting results. NIRS focuses on changes in the absorption spectrum of oxy hemoglobin in cortical lesions approximately 2.5~3 cm deep in the skin [5]. Accordingly, extracranial contaminations due to bony thickness and the cerebrospinal fluid layer of the cortex may distort the penetration of near-infrared light [23]. Okada et al. [24] reported that the partial optical path length is mainly affected by the inner skull surface when using a source-detector spacing of 30 mm in NIRS. In addition, the degree of perfusion in skin, blood supply to dura mater, or abnormal venous drainage into adjacent sinus also affect the NIRS measurements. Therefore, we excluded patients with SAH who underwent craniotomy and surgical clipping, and those with SAH concomitant with other cerebrovascular diseases such as moyamoya disease and arteriovenous malformation. 

Acute diagnosis of DCI and prompt intervention are critical factors determining favorable clinical outcomes following SAH. DCI diagnosis via consecutive steps is essential to rule out other diseases such as re-bleeding, seizures, hydrocephalus, infection, or electrolyte imbalances associated with neurologic decline. Accordingly, the diagnostic period varies with physicians’ expertise and radiology facilities. Therefore, a simple and robust method is required in the neurointensive care unit. TCD is usually used to screen cerebral vasospasm contributing to DCI. Nevertheless, inadequate temporal bone windows are a cause for concern. In our cohort, the TCD data of six patients (11.5%) was inadequate. In addition, compared with NIRS changes, TCD velocities appeared to show poor diagnostic performance, although the difference was not statistically significant (AUROC difference = 0.044, *p* = 0.571). Therefore, further comparative studies are required to evaluate the diagnostic efficacy of NIRS and TCD in a large cohort of SAH patients. 

In this study, we used a two-channel NIRS applied on both the frontal areas. The two-channel NIRS can be used to evaluate regional rSO2 on the watershed territories between the anterior cerebral artery and the middle cerebral artery (MCA). Therefore, it is difficult for two-channel NIRS to identify focal and distal MCA vasospasm or vasospasm of the posterior circulation. Mutoh et al. [25] used a 4-channel NIRS during the intra-arterial (IA) infusion of fasudil hydrochloride as vasospasm therapy, and demonstrated that rSO2 fluctuated in the vasospasm territory immediately after the IA infusion, followed by a gradual increase after selective infusion in the focal M1 segment of the MCA. The authors concluded that NIRS can be used to monitor real-time hemodynamic changes in a specific brain region during IA infusion to treat cerebral vasospasm. Since NIRS can be used to detect rSO2 changes where the sensor is placed [25], additional studies investigating the diagnostic accuracy of two-channel NIRS based on the frontal and temporal areas are necessary to assess focal and distal MCA vasospasm. 

Our findings may have been underpowered due to the small sample size, even though this study was the largest series reported to date. Second, the study only enrolled SAH patients who underwent coil embolization. Accordingly, the cutoff point may not be appropriate for SAH patients with surgical clipping. Third, no significant difference was found between NIRS and TCD for the detection of vasospasm-related DCI, although NIRS yielded better diagnostic accuracy with a difference in AUROC area of 0.044. Therefore, high-quality randomized controlled trials are required to evaluate the comparative diagnostic accuracy of NIRS and TCD.

## 5. Conclusions

NIRS represents a feasible option for the real-time detection of DCI via continuous rSO2 monitoring. Further comparative studies involving a large cohort of patients with SAH are needed to determine the optimal cutoff points for DCI detection.

## Figures and Tables

**Figure 1 jcm-09-01595-f001:**
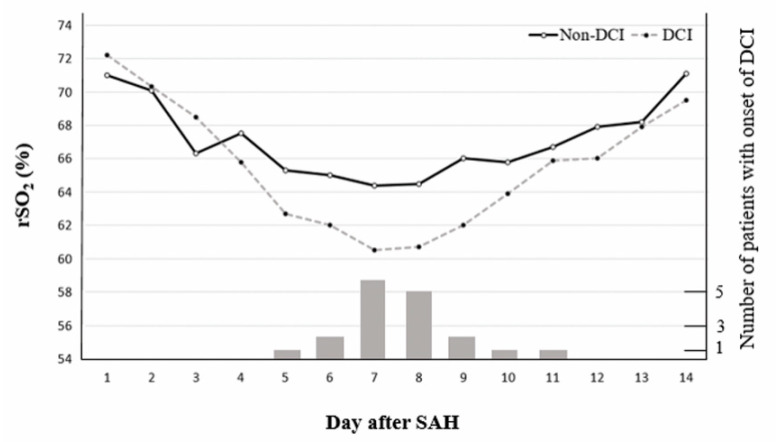
Chronological changes in cerebral regional oxygen saturation (rSO2) of near infrared spectroscopy (NIRS) according to delayed cerebral ischemia (DCI) and the number of patients showing the onset of DCI following subarachnoid hemorrhage (SAH).

**Figure 2 jcm-09-01595-f002:**
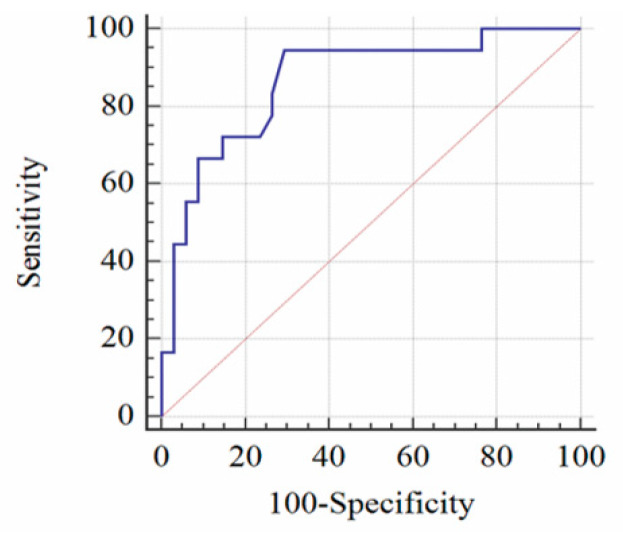
The area under the receiver operator characteristic curve is 0.865. Reduction in cerebral regional oxygen saturation (rSO2) >12.7% revealed a sensitivity of 94.44% (95% CI: 72.7–99.9%) and a specificity of 70.59% (95% CI: 52.5–84.9%). CI, confidence interval.

**Figure 3 jcm-09-01595-f003:**
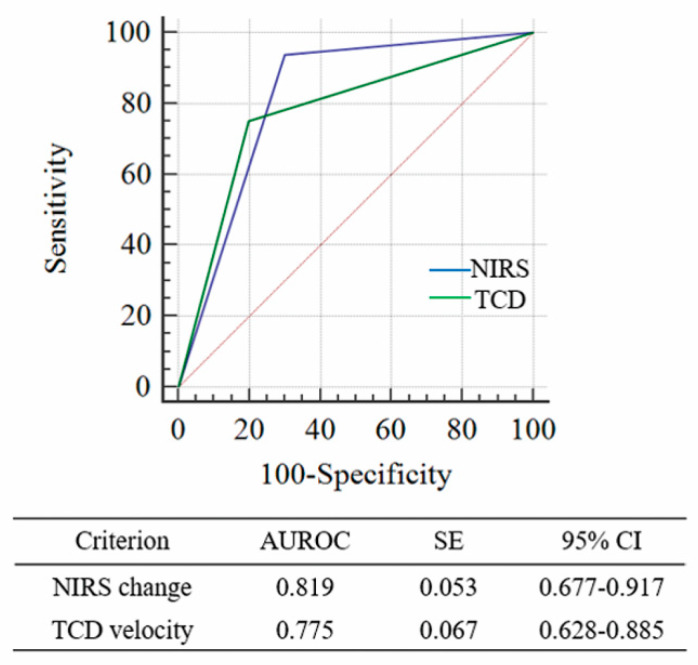
A comparison of receiver operating characteristic (ROC) curves involving NIRS changes and TCD velocity for the detection of severe vasospasm via cerebral angiography. The difference between AUROC curves was 0.044 (95% CI: −0.108 to 0.195; *p* = 0.571). AUROC, area under the ROC curve; CI, confidence interval; SE, standard error.

**Table 1 jcm-09-01595-t001:** Baseline characteristics of patients with subarachnoid hemorrhage enrolled in this study.

Variables	Non-DCI (*n* = 34)	DCI (*n* = 18)	*p*-Value
**Clinical findings**			
Female	18 (52.9%)	8 (44.4%)	0.564
Age, years	62.5 ± 10.6	58.6 ± 10.7	0.210
Hypertension	9 (26.5%)	5 (27.8%)	0.920
Diabetes mellitus	3 (8.8%)	3 (16.7%)	0.404
Hyperlipidemia	8 (23.5%)	3 (16.7%)	0.568
Smoking	5 (14.7%)	5 (27.8%)	0.260
H-H grade IV and V	12 (35.3%)	11 (61.1%)	0.077
**Laboratory finding**			
Hemoglobin	11.6 ± 1.1	11.2 ± 0.8	0.216
SaO_2_ (%)	95.5 ± 1.6	94.9 ± 2.5	0.384
MAP (mmHg)	92.6 ± 4.3	100.7 ± 5.4	<0.001
Heart rate (BPM)	90.4 ± 9.3	94.5 ± 8.4	0.124
**Radiologic findings**			
Anterior location	29 (85.3%)	16 (88.9%)	0.721
Size (mm)	5.4 ± 1.4	5.8 ± 1.2	0.311
Fisher grade 3 and 4	12 (35.3%)	13 (72.2%)	0.012
**Treatment outcome**			
Chemical angioplasty	-	7 (38.9%)	-
Poor outcome	8 (23.5%)	9 (50.0%)	0.055

BPM, beats per minute; DCI, delayed cerebral ischemia; H-H, Hunt and Hess; MAP, mean arterial pressure. Data are expressed as the numbers of subjects (percentage) representing discrete and categorical variables in the form of mean ± standard deviation.

**Table 2 jcm-09-01595-t002:** A linear mixed-effects model predicting NIRS parameters in patients with subarachnoid hemorrhage.

Variables	Standardized Estimate (95% CI)	SE	*p*-Value
H-H grade IV and V (vs. I, II, and III)	−1.76 (−2.75 to −0.76)	−1.76	<0.001
Fisher grade 3 and 4 (vs. 1 and 2)	−0.58 (−1.63 to 0.46)	0.53	0.272
MAP	0.04 (−0.04 to 0.11)	0.04	0.307
DCI	−1.46 (−2.47 to −0.46)	0.51	0.004

H-H: Hunt and Hess; MAP, mean arterial pressure; DCI, delayed cerebral ischemia; CI, confidence interval; SE, standard error.

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
