# Peer review of "Monitoring of Delayed Cerebral Ischemia in Patients with Subarachnoid Hemorrhage via Near-Infrared Spectroscopy"

_jcm, 2020, doi:10.3390/jcm9051595_

Round 1
Reviewer 1 Report
Non-invasive NIRS monitoring of hemoglobin saturation in cererbral tissue may provide important additional information for treatment decisions in aSAH patients developing DCI. The authors present a series of 52 endovascularly treated aSAH patients with continous rSO2 monitoring with two frontal NIRS patches over 14 days and define a threshold value of rSO2 decrease to trigger further diagnostics/treatment in DCI patients. Additionally, they compare the diagnostic accuracy of NIRS and TCD for detection of DCI. In general, this study provides valuable data for clinicians treating aSAH patients.
Suggestions:
Major:
- Please describe the temporal course of rSO2 changes and the development of DCI: Did rSO2 changes preceed clinical or radiological diagnosis of DCI?
Minor:
- It would be of interested to know the proportion of patients which were intubated and ventilated at the time point of DCI detection.
- Please describe the institutional standard monitoring and treatment protocol for aSAH patients (monitoring for intubated patients; rescue therapies etc.)
- Any data available from invasive neuromonitoring (ptiO2, microdialysis) and/or rCBF data from CT/MRI would be of interest at time point of DCI detection.
- Please discuss the potential "extracranial contamination" of non-invasive NIRS measurements (perfusion of skin, bone, dura supplied by external carotid artery).
- Please report technical issues during the continous monitoring period. It seems to be unrealistic to have continous monitoring data over 14 days in every patient. Drop-outs?
Author Response
Comments from the reviewer 1:
Major comment 1: Please describe the temporal course of rSO2 changes and the development of DCI: Did rSO2 changes precede clinical or radiological diagnosis of DCI?
Answer: Thank you for the comments. In this study, we tested the hypothesis that NIRS can be used to detect delayed cerebral ischemia (DCI) due to cerebral vasospasm. Most patients with DCI showed a progressive decline in rSO2 levels 3 days after ictus. Changes in rSO2 preceded the clinical symptoms of DCI. In particular, abrupt declines in rSO2 levels were associated with DCI development. We have further described the serial changes in rSO2 and the number of patients with different days of DCI in the supplemental figure legend.
Supplemental Figure: Data of 18 patients with delayed cerebral ischemia (DCI). Serial changes in cerebral regional oxygen saturation (rSO2) and the number of patients with different days of DCI onset are shown. Serial rSO2 changes and the day of DCI onset for each patient are presented in different colors. The changes in rSO2 are clinically relevant to identify the onset of DCI.
Minor comments 1~3:
- It would be of interested to know the proportion of patients which were intubated and ventilated at the time point of DCI detection.
- Please describe the institutional standard monitoring and treatment protocol for aSAH patients (monitoring for intubated patients; rescue therapies etc.)
- Any data available from invasive neuromonitoring (ptiO2, microdialysis) and/or rCBF data from CT/MRI would be of interest at time point of DCI detection.
Answer: Twenty-five patients were intubated after coil embolization and 11 of them were diagnosed with DCI. Potential complications associated with SAH usually occur in the first two weeks after ictus. Weir et al. (Reference 1, below) reported that cerebral vasospasm starts approximately on day 3, peaks on day 7, and resolves after two weeks following SAH. Thus, we monitored patients at least two weeks after coiling in the intensive care unit (References 2 and 3 below). Compared with good-grade SAH, patients with poor-grade SAH are challenge for the detection of DCI signs via neurological examination due to decreased consciousness levels (Reference 4, below). In addition, poor-grade SAH patients need frequent sedation to control increased intracranial pressure and to facilitate respiratory ventilation (Reference 2, below). Therefore, we routinely performed CT-angiography on days 3, 7 and day 14 to assess the degree of vasospasm after surgery. Further, TCD was performed daily to monitor cerebral flow velocities attributable to cerebral vasospasm. If DCI-related symptoms were suspected, such as new focal neurologic deficits or decreased GCS score of at least 2 points (Reference 2 below) with increased TCD velocity, we performed a CT scan and subsequent cerebral angiography for early intervention after excluding possible causes such as rebleeding and hydrocephalus. CT perfusion is a reliable tool for the detection of cerebral vasospasm and used widely in clinical settings. Greenberg et al. (Reference 5 below) reported that CT perfusion revealed a pooled sensitivity of 74.1% (95% CI: 58.7%-86.2%) and a specificity of 93.0% (95% CI: 79.6%-98.7%) for the diagnosis of vasospasm. Killeen et al. (Reference 6 below) also showed significant differences in cerebral blood flow between DCI (29.4 mL/100 g/min) and non-DCI patients (40.5 mL/100 g/min). However, the relative value of CT perfusion can be misleading in patients with diffuse vasospasm and vasospasm of vertebrobasilar artery (References 7 and 8 below). In addition, the results of CT perfusion are affected by extracranial carotid stenosis, proximal intracranial stenosis and cardiac output (Reference 7 below). The timing of endovascular intervention for DCI due to severe vasospasm was not undetermined. Although Hejcl et al. (Reference 9, below) demonstrated good angiographic outcome after intra-arterial chemical angioplasty, identification of vasospasm and subsequent chemical angioplasty in poor-grade SAH patients who were sedative or comatose were still difficult under real clinical settings (Reference 10, below). In such cases, microdialysis or direct brain tissue oxygenation is useful as an adjunct diagnostic tool for DCI due to vasospasm. However, their use is limited in the intensive care unit in Korea because the invasive procedures are not reimbursed by the National Health Insurance. Therefore, we prefer CTA and angiography for poor-grade SAH patients who are intubated, considering endovascular intervention at the time of angiography. If contraindicated for chemical angioplasty, hypertensive treatment is administered for reversal of cerebral blood flow. We have included monitoring and treatment protocol and the proportion of intubated patients in the Methods and Results section (page 2, line 61-63, 72-78).
References
- Weir B, Grace M, Hansen J, et al. Time course of vasospasm in man. J Neurosurg. 1978; 48:173-8.
- de Oliveira Manoel AL, Goffi A, Marotta TR, et al. The critical care management of poor-grade subarachnoid haemorrhage. Crit Care. 2016; 23:21
- Al-Khindi T, Macdonald RL, Schweizer TA, et al. Cognitive and functional outcome after aneurysmal subarachnoid hemorrhage. Stroke. 2010; 41: e519-36.
- Raimund Helbok, Pedro Kurtz, Michael J,et al. Effects of the neurological wake-up test on clinical examination, intracranial pressure, brain metabolism and brain tissue oxygenation in severely brain-injured patients. Crit Care. 2012; 16: R226.
- Greenberg ED, Gold R, Reichman M,et al. Diagnostic accuracy of CT angiography and CT perfusion for cerebral vasospasm: a meta-analysis. AJNR Am J Neuroradiol. 2010; 31:1853-60.
- Killeen RP, Mushlin AI, Johnson CE, et al. Comparison of CT perfusion and digital subtraction angiography in the evaluation of delayed cerebral ischemia. Acad Radiol. 2011;18:1094-100.
- Bruder, L. Velly, JL. Codaccioni. Modern Approach to SAH in Intensive Care Unit (ICU). Interv Neuroradiol. 2008; 14(Suppl 1): 13–16.
- Lui YW, Tang ER, Allmendinger AM, et al. Evaluation of CT perfusion in the setting of cerebral ischemia: patterns and pitfalls. AJNR Am J Neuroradiol. 2010; ;31:1552-63.
- Hejčl A, Cihlář F5, Smolka V, et al. Chemical angioplasty with spasmolytics for vasospasm after subarachnoid hemorrhage. Acta Neurochir (Wien). 2017; 159:713-720.
- Menon G.Vasospasm following aneurysmal subarachnoid hemorrhage: The search for the elusive wonder-drug. Neurol India. 2018; 66:423-425.
Minor comment 4: Please discuss the potential "extracranial contamination" of non-invasive NIRS measurements (perfusion of skin, bone, dura supplied by external carotid artery).
Answer: Per your recommendation, we have revised the Discussion as follows:
NIRS focuses on changes in the absorption spectrum of oxy hemoglobin in cortical lesions approximately 2.5~3 cm deep in the skin (Reference 1 below). Accordingly, extracranial contamination due to bony thickness and the layer of cerebrospinal fluid in the cortex may distort the penetration of near-infrared light (Reference 2 below). Okada et al. (Reference 3 below) reported that the partial optical path length is mainly affected by the inner skull surface when using a source-detector spacing of 30 mm in NIRS. In addition, the degree of perfusion in skin, and the blood supply to dura mater or abnormal venous drainage into adjacent sinus can also affect the NIRS measurements. Therefore, we excluded SAH patients who underwent craniotomy and surgical clipping and those with other concomitant cerebrovascular diseases such as moyamoya disease and arteriovenous malformation. We have discussed the potential extracranial contamination of NIRS (page 6, line 174-181).
References
- Hoffman GM, Ghanayem NS, Tweddell JS. Noninvasive assessment of cardiac output. Semin Thorac Cardiovasc Surg Pediatr Card Surg Annu. 2005:12-21
- Poon WS, Wong GK, Ng SC. The quantitative time-resolved near infrared spectroscopy (tr-nirs) for bedside cerebrohemodynamic monitoring after aneurysmal subarachnoid hemorrhage: Can we predict delayed neurological deficits? World Neurosurg. 2010;73:465-466
- Okada E, Delpy DT. Near-infrared light propagation in an adult head model. I. Modeling of low-level scattering in the cerebrospinal fluid layer. Appl Opt. 2003; 42:2906-14.
Minor comment 5: Please report technical issues during the continous monitoring period. It seems to be unrealistic to have continous monitoring data over 14 days in every patient. Drop-outs?
Answer: NIRS measurement is affected by soft tissue swelling, hair or pneumocephalus. Thus, patients treated with craniotomy may have distorted results. For this reason, we enrolled SAH patients who underwent coil embolization, and excluded those who underwent surgical clipping. Therefore, we did not encounter technical issues during the monitoring.
Per your comments, it is not easy to use NIRS in all SAH patients until two weeks after ictus in the real-world critical care settings. In addition, good-grade SAH patients complained about the NIRS pad on their forehead. In this study, we demonstrated the feasibility of NIRS for monitoring vasospasm-related DCI to develop a neuromonitoring reference standard after coil embolization. In Korea, modalities such as NIRS as well as microdialysis and direct brain tissue oxygenation are not reimbursed by the National Health Insurance. Therefore, we used NIRS to cover the vasospasm period of two weeks after ictus. However, per your comments, we do not believe that every patient requires two-week monitoring of NIRS following SAH. We think that only SAH patients who are at high risk of DCI should be monitored during the vasospasm period. Therefore, a further study is required to investigate the role of NIRS in the detection of DCI and clinical outcomes in a large cohort of SAH population.

Reviewer 2 Report
J Clin Med: near-infrared spectroscopy for DCI
The authors evaluated NIRS for the detection of post-hemorrhagic vasospasm.
In general, this is an interesting and clinically relevant topic. The paper needs significant improvement.
A native English-speaking person with a medical background should revise the text (e.g., “24-hour monitoring is not allowed in the intensive care setting”??).
The fact that only patients after endovascular aneurysm treatment were enrolled should be mentioned in the abstract.
Given the risks of vasospasm and the concerning clinical sequelae, the risks associated with the transfer of the patient from the ICU to any examination can be neglected. This is a weak argument, which should be deleted since it is misleading. There is no reason to avoid CT Perfusion and/or DSA in patients at risk of severe vasospasm after aneurysmal SAH – unless there is a well-established point-of-care monitoring modality available.
The finding in the ISAT data that clipping was more frequently associated with DCI than coiling is a poor argument for the exclusion of clipped patients from the presented study. It is reasonable to limit the evaluation of NIRS to non-surgical patients, but on based on this argument.
Many statements are incomprehensible:
“The primary endpoint involved association between NIRS and DCI development following SAH…”
Did all patients have a DSA examination? Only those with vasospasm? I understand that only in the case of suspected DCI a DSA was performed – triggered by a clinical deterioration. How about unconscious patients? This is the most critical subgroup.
The focus on DCI instead of vasospasm is a problem. Vasospasm is the key phenomenon, which may or may not result in DCI. I therefore propose to restructure the results as follows:
From N=52 enrolled patients, X were identified as developing vasospasm and Y eventually developed DCI. From these X (vasospasm) patients (as defined by clinical deterioration and DSA showing vasospasm), TCD was positive in # patients and NIRS was positive in ## patients. I recommend to undertake all possible efforts to clarify and simplify the results.
The sensitivity of TCD for severe vasospasm was poor (75% after excluding patients with no bone window), as reported before. NIRS was not better in this regard - and THIS is the key finding of this paper, which deserves to be communicated.
We need a method to identify posthemorrhagic vasospasm prior to DCI in clipped and coiled patients and without daily CTP or DSA. This is, unfortunately, not NIRS.
Author Response
Comments from the reviewer 2:
Comment 1: A native English-speaking person with a medical background should revise the text (e.g., “24-hour monitoring is not allowed in the intensive care setting”??).
Answer: Per your comments, we have retained the services of a professional editing company employing native speakers.
Comment 2: The fact that only patients after endovascular aneurysm treatment were enrolled should be mentioned in the abstract.
Answer: Per your comment, we have revised the abstract as follows:
Fifty-two patients treated with endovascular coil embolization were enrolled, including 18 with DCI (34.6%) and 34 without DCI (65.4%).
Comment 3: Given the risks of vasospasm and the concerning clinical sequelae, the risks associated with the transfer of the patient from the ICU to any examination can be neglected. This is a weak argument, which should be deleted since it is misleading. There is no reason to avoid CT Perfusion and/or DSA in patients at risk of severe vasospasm after aneurysmal SAH – unless there is a well-established point-of-care monitoring modality available.
Answer: Thank you for your comments. Accordingly, we have deleted the sentences from the introduction.
Comment 4: The finding in the ISAT data that clipping was more frequently associated with DCI than coiling is a poor argument for the exclusion of clipped patients from the presented study. It is reasonable to limit the evaluation of NIRS to non-surgical patients, but on based on this argument.
Answer: Per your suggestion, we have deleted the sentences. In addition, we have added an explantion of the inclusion criteria as follows: Because SAH patients undergoing surgical clipping had air, fluid or hemorrhage in the craniotomy sites which can distort NIRS measurement, we only placed the NIRS sensor pad in SAH patients treated with coil embolization (page 2, line 61-63).
Comment 5: Many statements are incomprehensible: “The primary endpoint involved association between NIRS and DCI development following SAH…”
Answer: Per your recommendation, we have retained the services of a native speaker to edit the manuscript. The sentences are revised as follows:
The primary outcome was the association between NIRS and DCI following SAH, suggesting the optimal cutoff value of rSO2 reduction for identification of DCI (page 2, line 67-69).
Comments 6 and 9:
- Did all patients have a DSA examination? Only those with vasospasm? I understand that only in the case of suspected DCI a DSA was performed – triggered by a clinical deterioration. How about unconscious patients? This is the most critical subgroup.
- We need a method to identify posthemorrhagic vasospasm prior to DCI in clipped and coiled patients and without daily CTP or DSA. This is, unfortunately, not NIRS.
Answer: DSA was performed in cases with suspected DCI defined as newly developed focal neurologic deficits or decreased GCS scores of at least 2 points concomitant with severe angiographic vasospasm. Compared with conscious patients, unconsicous patients presenting with poor-grade SAH is a challenge for the detection of DCI based on neurologic examination, due to decreased consciousness levels (Reference 1, below). In addition, poor-grade SAH patients need frequent sedation to control the increased intracranial pressure (IICP) and to facilitate respiratory ventilation (Reference 2, below). Therefore, we routinely performed CT-angiography on days 3, 7 and 14 to assess the degree of vasospasm after surgery. Further, TCD was performed daily to monitor the cerebral flow velocity due to cerebral vasospasm. If DCI-related symptoms are suspected such as new focal neurologic deficits or decreased GCS score of at least 2 points (Reference 2 below) with increased TCD velocity, we performed a CT scan and subsequent cerebral angiography for early intervention after excluding possible causes such as rebleeding and hydrocephalus. CT perfusion is a reliable tool for the detection of cerebral vasospasm and is widely used in clinical settings. Greenberg et al. (Reference 3 below) reported that CT perfusion revealed a pooled sensitivity of 74.1% (95% CI: 58.7%-86.2%) and a specificity of 93.0% (95% CI: 79.6%-98.7%) for the diagnosis of vasospasm. Killeen et al. (Reference 4 below) also showed significant variation in cerebral blood flow between DCI (29.4 mL/100 g/min) and non-DCI patients (40.5 mL/100 g/min). However, the relative value of CT perfusion can be misleading in patients with diffuse vasospasm and vasospasm of vertebrobasilar artery (References 5 and 6 below). In addition, the results of CT perfusion are affected by extracranial carotid stenosis, proximal intracranial stenosis and cardiac output (Reference 5 below). The timing of endovascular intervention for DCI due to severe vasospasm remain undetermined. Although Hejcl et al. (Reference 7, below) demonstrated good angiographic outcome after intra-arterial chemical angioplasty, detection of vasospasm and subsequent chemical angioplasty in poor-grade SAH patients who are sedated or comatose are still a challenge in clinical settings (Reference 8, below). In such cases, microdialysis or direct brain tissue oxygenation is useful as an adjunct diagnostic tool for DCI due to vasospasm. However, their use is limited in the intensive care unit in Korea because the invasive procedures are not reimbursed by the National Health Insurance. Therefore, we prefer CTA and angiography for poor-grade SAH patients who are intubated, considering endovascular intervention at the time of angiography. If contraindicated for chemical angioplasty, hypertensive treatment is administered for reversal of cerebral blood flow. We have included monitoring and treatment protocols and the proportion of intubated patients in the Methods and Result section (page 2, line 72-78; page 3, line 107-108).
References
- Raimund Helbok, Pedro Kurtz, Michael J,et al. Effects of the neurological wake-up test on clinical examination, intracranial pressure, brain metabolism and brain tissue oxygenation in severely brain-injured patients. Crit Care. 2012; 16: R226.
- de Oliveira Manoel AL, Goffi A, Marotta TR, et al. The critical care management of poor-grade subarachnoid haemorrhage. Crit Care. 2016; 23:21
- Greenberg ED, Gold R, Reichman M,et al. Diagnostic accuracy of CT angiography and CT perfusion for cerebral vasospasm: a meta-analysis. AJNR Am J Neuroradiol. 2010; 31:1853-60.
- Killeen RP, Mushlin AI, Johnson CE, et al. Comparison of CT perfusion and digital subtraction angiography in the evaluation of delayed cerebral ischemia. Acad Radiol. 2011;18:1094-100.
- Bruder, L. Velly, JL. Codaccioni. Modern Approach to SAH in Intensive Care Unit (ICU). Interv Neuroradiol. 2008; 14(Suppl 1): 13–16.
- Lui YW, Tang ER, Allmendinger AM, et al. Evaluation of CT perfusion in the setting of cerebral ischemia: patterns and pitfalls. AJNR Am J Neuroradiol. 2010; ;31:1552-63.
- Hejčl A, Cihlář F5, Smolka V, et al. Chemical angioplasty with spasmolytics for vasospasm after subarachnoid hemorrhage. Acta Neurochir (Wien). 2017; 159:713-720.
- Menon G.Vasospasm following aneurysmal subarachnoid hemorrhage: The search for the elusive wonder-drug. Neurol India. 2018; 66:423-425.
Comment 7: The focus on DCI instead of vasospasm is a problem. Vasospasm is the key phenomenon, which may or may not result in DCI. I therefore propose to restructure the results as follows:
From N=52 enrolled patients, X were identified as developing vasospasm and Y eventually developed DCI. From these X (vasospasm) patients (as defined by clinical deterioration and DSA showing vasospasm), TCD was positive in # patients and NIRS was positive in ## patients. I recommend to undertake all possible efforts to clarify and simplify the results.
Answer: Thank you for comments on our manuscript. We agree with your comments, and accordingly, DCI is a multifactorial disease entity based on a single episode of angiographic cerebral vasospasm, cortical spreading ischemia and microthrombosis or their combination (Reference 1 below), although angigraphic vasospasm has been mostly associated with DCI. Without multimodality monitoring, it is not easy to determine the pathophysiology of DCI. However, as mentioned above, invasive monitoring is not allowed in critical care setting in Korea due to lack of reimbursement. In addition, diagnosis of DCI in patients presenting with poor-grade SAH is challenging due to their decreased level of consicousness and frequent use of sedative drgus to control IICP or to facilitate respiratory ventilation. Therefore, we defined DCI as newly developed focal neurologic deficits or decreased GCS score of at least 2 points concomitant with severe angiographic vasospasm. Carreta et al. (Reference 2, below) defined DCI as clinical detrioration with or without a new infarction on CT that was suspected as a cerebral vasospasm. Neverthless, we admitted that some DCI patients with cortical spreading ischemia and microthrombosis may have been neglected in this study.
References
- de Oliveira Manoel AL, Goffi A, Marotta TR, et al. The critical care management of poor-grade subarachnoid haemorrhage. Crit Care. 2016; 23:21
- Carrera E1, Schmidt JM, Oddo M, et al. Transcranial Doppler for predicting delayed cerebral ischemia after subarachnoid hemorrhage. Neurosurgery. 2009;65:316-23.
Comment 8: The sensitivity of TCD for severe vasospasm was poor (75% after excluding patients with no bone window), as reported before. NIRS was not better in this regard - and THIS is the key finding of this paper, which deserves to be communicated.
Answer: Thank you for your comemnts. Kumar et al. (Reference 1, below) reported a sensitivity of 90% (95% CI: 77%-96%) and a specificity of 71% (95% CI: 51%-84%) for TCD diagnosis of cerebral vasospasm. However, heteogeneity (I2 > 50%) across the studies due to the differences in DCI definition and its diagnosis, and the cutoff value of mean flow velocites for vasospasm, can lead to different interpretations. In this study, we foucsed on DCI due to severe angiographic vasospasm. Severe vasospasm detected via TCD was defined by a mean flow velocity higher than 200 cm/s in the middle cerebral artery (MCA) or 85 cm/s in the basilar artery (Refrences 2 and 3, below). Samagh et al. (Reference 4 below) reproted that very high MCA flow velocities (>200 cm/s) or low velocites (<120 cm/s) can be used to predict clinically significant vasospassm reliably. Nevertheless, we found no significant difference between NIRS and TCD for the detection of vasospasm-related DCI, although NIRS yielded better diagnostic accuracy with a difference in AUROC area of 0.044. Therefore, high-quality randomized controlled trials are needed to evaluate the comparative diagnostic accuracy of NIRS and TCD. We have discussed the limiation in the manuscript (page 6, line 208-211).
References
- Kumar G, Shahripour RB, Harrigan MR. Vasospasm on transcranial Doppler is predictive of delayed cerebral ischemia in aneurysmal subarachnoid hemorrhage: a systematic review and meta-analysis. J Neurosurg. 2016;124:1257-64.
- Vora YY, Suarez-Almazor M, Steinke DE, Martin ML, Findlay JM. Role of transcranial doppler monitoring in the diagnosis of cerebral vasospasm after subarachnoid hemorrhage. Neurosurgery. 1999;44:1237-1247; discussion 1247-1238
- Samagh N, Bhagat H, Jangra K. Monitoring cerebral vasospasm: How much can we rely on transcranial doppler. J Anaesthesiol Clin Pharmacol. 2019;35:12-18
